# Copy Number Profiles of Prostate Cancer in Men of Middle Eastern Ancestry

**DOI:** 10.3390/cancers13102363

**Published:** 2021-05-14

**Authors:** Alia Albawardi, Julie Livingstone, Saeeda Almarzooqi, Nallasivam Palanisamy, Kathleen E. Houlahan, Aktham Adnan Ahmad Awwad, Ramy A. Abdelsalam, Paul C. Boutros, Tarek A. Bismar

**Affiliations:** 1Tawam Hospital, Abu Dhabi P.O. Box 15258, United Arab Emirates; Alia.albawardi@uaeu.ac.ae (A.A.); Saeeda.almarzooqi@uaeu.ac.ae (S.A.); aadnan@union71.ae (A.A.A.A.); 2Pathology College of Medicine & Health Sciences, United Arab Emirates University, Al Ain, Abu Dhabi P.O. Box 15551, United Arab Emirates; 3Departments of Human Genetics, University of California, Los Angeles, CA 94607, USA; JLivingstone@mednet.ucla.edu (J.L.); KHoulahan@mednet.ucla.edu (K.E.H.); PBoutros@mednet.ucla.edu (P.C.B.); 4Jonsson Comprehensive Cancer Centre, University of California, Los Angeles, CA 94607, USA; 5Institute for Precision Health, University of California, Los Angeles, CA 94607, USA; 6Department of Urology, Vattikuti Urology Institute, Henry Ford Health System Detroit, Detroit, MI 48202, USA; NPALANI1@hfhs.org; 7Department of Medical Biophysics, University of Toronto, Toronto, ON M5G 1L7, Canada; 8Department of Pathology and Laboratory Medicine, University of Calgary-Cumming School of Medicine and Alberta Precision Labs, Calgary, AB T2N 4N1, Canada; Ramyabdelsalam666@gmail.com; 9Department of Pathology, Mansoura University, Mansoura 35516, Egypt; 10Department of Pharmacology & Toxicology, University of Toronto, Toronto, ON M5S 1A8, Canada; 11Department of Urology, University of California, Los Angeles, CA 94607, USA; 12Departments of Oncology, Biochemistry and Molecular Biology, University of Calgary-Cumming School of Medicine, Calgary, AB T2N 4N1, Canada; 13Arnie Charbonneau Cancer Institute and Tom Baker Cancer Center, Calgary, AB T2N 4N1, Canada; 14Alberta Precision Labs, Rockyview Hospital Laboratory, Department of Pathology & Laboratory Medicine, University of Calgary Cumming School of Medicine, 7007-14th Street SW, Calgary, AB T2V 1P9, Canada

**Keywords:** prostate cancer, middle eastern ancestry, copy number aberrations

## Abstract

**Simple Summary:**

Prostate cancer is the most commonly diagnosed non-skin malignancy in men. Numerous studies have been undertaken to explore the role that genomics plays in prostate cancer initiation and progression. Most of this genomic data comes tumors arising in men with European or Asian ancestry, leaving other ancestry groups understudied. To fill this gap, we investigated the differences in copy number aberrations between prostate cancers arising in men of Middle Eastern ethnicity and those of European, African, or East Asian ethnicities in the hope of better understanding the incidence and risk of prostate cancer in different populations. We identified ancestry-specific gains and deletions, as well as differences in overall genomic instability between ancestry groups. This confirms that ancestry should be considered when investigating and characterizing biomarkers and molecular signatures relative to disease progression, prognosis, and potentially therapeutic targeting.

**Abstract:**

Our knowledge of prostate cancer (PCa) genomics mainly reflects European (EUR) and Asian (ASN) populations. Our understanding of the influence of Middle Eastern (ME) and African (AFR) ancestry on the mutational profiles of prostate cancer is limited. To characterize genomic differences between ME, EUR, ASN, and AFR ancestry, fluorescent in situ hybridization (FISH) studies for *NKX3-1* deletion and MYC amplification were carried out on 42 tumors arising in individuals of ME ancestry. These were supplemented by analysis of genome-wide copy number profiles of 401 tumors of all ancestries. FISH results of *NKX3-1* and *MYC* were assessed in the ME cohort and compared to other ancestries. Gene level copy number aberrations (CNAs) for each sample were statistically compared between ancestry groups. *NKX3*-1 deletions by FISH were observed in 17/42 (17.5%) prostate tumors arising in men of ME ancestry, while *MYC* amplifications were only observed in 1/42 (2.3%). Using CNAs called from arrays, the incidence of *NKX3-1* deletions was significantly lower in ME vs. other ancestries (20% vs. 52%; *p* = 2.3 × 10^−3^). Across the genome, tumors arising in men of ME ancestry had fewer CNAs than those in men of other ancestries (*p* = 0.014). Additionally, the somatic amplification of 21 specific genes was more frequent in tumors arising in men of ME vs. EUR ancestry (two-sided proportion test; Q < 0.05). Those included amplifications in the glutathione S-transferase family on chromosome 1 (*GSTM1*, *GSTM2*, *GSTM5*) and the IQ motif-containing family on chromosome 3 (*IQCF1*, *IQCF2*, *IQCF13*, *IQCF4*, *IQCF5*, *IQCF6*). Larger studies investigating ME populations are warranted to confirm these observations.

## 1. Introduction

Prostate cancer (PCa) is the most commonly diagnosed non-skin malignancy in men. Risk factors include age [1], family history [2], diet, obesity, and ancestry [3]. Genome-wide association studies have identified common genetic polymorphisms on chromosome 8p and elsewhere to be a significant influence on PCa initiation and progression in populations of European ancestry [4]. Similarly, specific germline polymorphisms have been associated with prostate cancer risk in men of various ancestries [5,6].

In a similar way, several somatic genomic alterations differ in frequency between ancestries, including *TMPRSS2-ERG* genomic rearrangements, *SPINK1* overexpression, and *SPOP* single nucleotide variants (SNVs), supporting differences in prostate cancer risk and pathway progression [7,8]. The rate of *PTEN* loss, as assessed by IHC, is lower in men of ME ethnicity, and tumors with *PTEN* loss are not enriched for *ERG* loss, as documented in EUR cohorts [9].

Copy number aberrations (CNAs) are the main mutational drivers of prostate cancer, occurring early in evolution [10], being highly prognostic [11], driving subtype [12], and shaping the tumor transcriptome and proteome [13]. The large majority of prostate cancer genomic data are from men with European or Asian ancestry [14], leaving other ancestry groups understudied. To begin to fill this gap, we assembled a cohort of publicly available CNA data from 376 patients of diverse ancestries, and new data from 25 men of Middle Eastern (ME) ancestry. The ME cohort also included 42 samples assessed by FISH for *NKX3-1* deletion and *MYC* amplification.

## 2. Methods

### 2.1. Fluorescence In Situ Hybridization Patient Cohort

A retrospective cohort of 41 radical prostatectomies and one transurethral resection of the prostate from Tawam Hospital (Al Ain, United Arab Emirates), reflecting Middle Eastern men, was investigated by FISH for *NKX3-1* deletion and *MYC* amplification. All selected cases had histopathologic diagnosis of prostate adenocarcinoma. The original histopathologic classification was confirmed by one of the study pathologists (AA, SA, or TAB). Formalin-fixed, paraffin-embedded tissue sections were used for interphase FISH. Deparaffinized tissue was treated with 0.2 mol/L HCl for 10 min, 2 × SSC for 10 min at 80 °C and digested with Proteinase K (Invitrogen, Carlsbad, CA, USA) for 8 min. The tissues and BAC probes were co-denatured for 5 min at 94 °C and hybridized overnight at 37 °C. Post-hybridization washing was with 2 × SSC with 0.1% Tween 20 for 5 min, and fluorescent detection was done using anti-digoxigenin conjugated to fluorescein (Roche Applied Science, Indianapolis, IN, USA), and streptavidin conjugated to Alexa Fluor 594 (Invitrogen). Slides were counterstained and mounted in ProLong Gold Antifade Reagent with 4′,6-diamidino-2-phenylindole (Invitrogen). Slides were examined using a Leica DMRA fluorescence microscope (Leica, Deerfield, IL, USA) and imaged with a CCD camera using the CytoVision software system (Applied Imaging, Santa Clara, CA, USA). FISH signals were scored manually (X100 oil immersion) in morphologically intact and non-overlapping nuclei by pathologists, and a minimum of 100 cancer cells from each site were recorded. Cancer sites with very weak or no signals were recorded as insufficiently hybridized. All BACs were obtained from the BACPAC Genomics (Emeryville, CA, USA), and probe locations were verified by hybridization to metaphase spreads of normal peripheral lymphocytes. For detection of gene deletion or amplification, the following probes were used: for *NKX3-1*, RP11-325C22 (green), and for *MYC*, RP11-1136L8 (red). BAC DNA was isolated using a QIAFilter Maxi Prep kit (Qiagen, Valencia, CA, USA), and probes were synthesized using digoxigenin- or biotin-nick translation mixes (Roche Applied Science).

### 2.2. OncoScan SNP Array Patient Cohort

We assessed 25/42 of the FISH cohort cases that had sufficient tissue with OncoScan SNP microarrays (21 cases had both SNP and FISH data available), since some cores were not available for interpretation by FISH on the TMA, due to technical issues. Representative slide(s) from each case corresponding to the formalin-fixed paraffin-embedded block(s), were identified to obtain tissue cores for OncoScan SNP microarray profiling. For molecular analysis, approximately 8–10 punches of 1.5 mm were collected in Eppendorf tubes from regions with highest percentage of invasive carcinoma, non-tumoral, and involved regional lymph nodes if any.

Ethical approval was obtained from the Al Ain Medical District Human Research Ethics Committee # 15/118 (2015–4246). This study was supported by a grant from Sheikh Hamdan Bin Rashid Al Maktoum Award for Medical Sciences (MRG/35/2017). Patient selection, tissue collection, and sample processing for the 376 remaining non-ME samples were performed as described in a previous publication [12].

### 2.3. SNP Microarray Data Generation and CNA Calling

SNP microarrays were performed with 200 ng of DNA on Affymetrix OncoScan FFPE Express 3.0 arrays as previously described [12]. The genotypes of 217,439 SNPs were extracted from the OncoScan OSCHP files and converted to VCF format. Gene level CNAs for each patient were identified by overlapping copy number segments, with RefGene (2014-07-15) annotation, using BEDTools (v2.17.0) [15]. Percent genome altered (PGA) was calculated for each sample by dividing the number of base pairs that were involved in all copy number segments by the total length of the genome.

### 2.4. Inferring Ancestry

The HGDP-CEPH dataset (*n* = 1042) was downloaded [16], subsetted to SNPs that overlap with the OncoScan SNP array (*n* = 63,320), and converted to VCF format. HGDP-CEPH VCFs were merged with the OncoScan cohort using VCFtools [17]. SNPs in linkage equilibrium with each other were pruned using the -indep command in PLINK [18]. Principal component analysis (PCA) was implemented using PLINK 1.9 (--pca) in the entire dataset as well as within the European population and Middle Eastern population separately.

The ADMIXTURE v 1.3.0 algorithm [19] was used to infer the ancestry of individuals based on the ancestry proportion given *k* ancestral populations with the PLINK BED file as input. Unsupervised ADMIXTURE analysis was run with k-fold cross validation with iterations of *k* from 2 to 8. This assignment was used to compare CNA profiles between ancestry populations.

### 2.5. Statistical Analysis

The specific statistical tests used are indicated in the figure legends or appropriate methods section and were performed within the R statistical environment (v3.3.1). Visualization in R was performed with the BPG package (v5.9.2) [20].

## 3. Results

To assess the CNA changes in prostate tumors from men with Middle Eastern ancestry, we first focused on two known prostate-cancer-associated genes, *NKX3-1* and *MYC*. Using FISH, we determined that 16% (7/42) of prostate tumors arising in men of Middle Eastern ancestry harbored an *NKX3-1* deletion. Similarly, an assessment by FISH detected that ~2.5% (1/42) of tumors arising in men of ME ancestry had a *MYC* amplification. Appendix A shows an example tumor with a *NKX3-1* deletion detected by FISH.

Next, to compare these intriguing results with different ancestral populations, we assembled a dataset of 401 patients with sporadic, localized, treatment-naive disease. Each patient in this cohort had whole genome copy-number profiling of the index lesion of their tumor. This cohort included 25 newly generated tumors from men of Middle Eastern ancestry (Table 1; Appendix A). Patients underwent either image-guided radiotherapy (IGRT) or surgery (radical prostatectomy), and the histologically most representative region was molecularly profiled. There was no difference in cellularity (two-way ANOVA; *p* = 0.15; Appendix A), as measured by ASCAT [21]. We observed a similar percentage of CNAs by SNP microarrays, where 20% (5/25) of tumors from men with Middle Eastern ancestry had a deletion of *NKX3-1*. By comparison, 54% (186/346) of tumors arising in men of European ancestry harbored a deletion of *NKX3-1* (*p* = 2.3 × 10^−3^). The rate of MYC amplification detection by SNP microarray was higher at 16% (4/25), and comparable to the 18% (65/346) in men of European ancestry (*p* = 0.94).

### 3.1. Genetic Ancestry Inference

To infer the genetic ancestry of the assembled dataset, we integrated our data with the Human Genome Diversity Panel (HGDP-CEPH), which contains samples from 51 different global populations [16]. To determine the ancestry fraction for each individual, we used the software ADMIXTURE [19] (Figure 1A; Appendix A). HGDP-CEPH samples previously assigned an ancestry group were not re-assigned, independent of the admixture results. Four primary admixture populations were apparent (Africa-Red; America-Yellow, East Asia-Green, and Oceania-Aquamarine). The other populations were a combination of these primary populations, including samples from men with Middle Eastern ancestry, who showed a combination of African, Central South Asian, and European genotypic features.

Three Middle Eastern samples in our dataset contained a larger than average proportion of the African primary admixture component (Appendix A). We performed principal component analysis (PCA) using 63,320 SNPs to investigate genetic diversity (Figure 1B). PC1 and PC2 explained 30% and 27% of genetic variance, respectively, with PC1 distinguishing East from West populations, while PC2 divided African/Middle Eastern populations from non-African populations. PCA of specific geographical regions showed sub-structures within the larger populations, including within specific regions of Europe and the Middle East (Figure 1C,D). As previously reported [22,23], Middle Eastern and European populations have divergent genetic ancestry.

### 3.2. Ancestry-Associated CNA Mutation Density

To explore differences in somatic CNA profiles between different ancestry groups, we compared the copy number profiles of tumors from men with Middle Eastern ancestry (*n* = 25), African ancestry (*n* = 14), East Asian ancestry (*n* =11), and European ancestry (*n* = 346; Figure 2A; Appendix A). Tumors identified as arising in men of Native American ancestry (*n* = 5) were removed due to the small sample size. All ancestry groups showed large inter-tumoral heterogeneity, where tumors from men with European ancestry had between 0–284 CNAs, tumors from men with African ancestry had 19–217 CNAs, tumors from men with East Asian ancestry had 4–209 CNAs, and tumors from men with Middle Eastern ancestry had 1–166 CNAs. 

The median tumor arising in men of European ancestry contained 18 amplifications and 29 deletions. Tumors arising in men of East Asian ancestry had a similar number of CNAs (20 amplifications + 33 deletions; *p* = 0.16; Man-Whitney test). By contrast, tumors arising in men of Middle Eastern ancestry had fewer CNAs (7 amplifications + 11 deletions; *p* = 0.014; Mann-Whitney test), and those arising in men of African ancestry had the most CNAs (45 amplifications + 27 deletions; *p* = 0.016; Mann-Whitney test). As expected, the smaller number of CNAs in tumors from men with Middle Eastern ancestry led to lower global genomic instability, measured as the percentage of genome altered (PGA; Figure 2B). Tumors in men with Middle Eastern ancestry had lower PGA (median = 1.93%) than those arising in men of European ancestry (median = 6.45%; *p* = 1.07 × 10^−2^; Mann-Whitney test). There was no difference in PGA between tumors from men with African ancestry despite the larger number of aberrations (median = 9.72%) or tumors from men with East Asian ancestry (median = 6.83%). 

### 3.3. Ancestry-Associated CNAs 

We assigned each tumor to a prostate cancer subtype based on their gene level CNA profile (Figure 2A,C) [11]. These subtypes (S1–S4) were defined by specific genomic aberrations (S1-chromosome 7 amplification; S2-8p deletion, 8q amplification; S3-8p deletion, 16p deletion; S4-quiet profile). All genetic ancestry groups had a large proportion of tumors in S4, indicative of quiet CNA profiles (Figure 2A). Despite the differences in global genomic instability, the proportion of tumor subtypes did not differ between the different genetic ancestry groups, at least at the statistical power afforded by the existing cohorts (Pearson’s Χ^2^ test; *p* = 0.16; Figure 2C).

Understanding the interplay between genetic ancestry and CNAs is important for ancestry-specific biomarker identification. To this end, we compared recurrent gene level CNAs in tumors of men with Middle Eastern, African, and East Asian ancestry with those of men with European ancestry (Appendix A). We overlapped CNA segments with gene annotation and compared the proportion of amplifications and deletions separately. Overall, when investigated globally, there were no differences in the proportion of amplifications and deletions in known prostate cancer driver genes across ancestry populations at this statistical power (two-sided proportion test; Q < 0.05; Appendix A; Appendix A).

Globally, 21 genes had more frequent amplifications in tumors from men with Middle Eastern ancestry compared to tumors from men with European ancestry (two-sided proportion test; Q < 0.05; Figure 2D). All of these amplifications occurred at a higher rate in tumors from men with Middle Eastern ancestry. These included amplifications in the glutathione S-transferase family of genes on chromosome 1 (*GSTM1*, *GSTM2*, *GSTM5*) and the IQ motif-containing family of genes on chromosome 3 (*IQCF1*, *IQCF2*, *IQCF13*, *IQCF4*, *IQCF5*, *IQCF6*). Interestingly, *PARP3*, an enzyme which is required for DNA repair and maintenance of genomic instability, was amplified in a higher proportion of tumors from men with Middle Eastern ancestry. We did not observe any differences in the proportion of deletions between these two populations (two-sided proportion test; Q < 0.05).

There were 34 genes in which the proportion of tumors with an amplification, and 40 genes in which the proportion of deletions, differed between tumors from men with African ancestry compared with tumors from men with European ancestry (two-sided proportion test; Q < 0.05; Figure 2D). These include amplifications in the keratin-associated protein 10 family of genes on chromosome 21 (*KRTAP10-1*-*KRTAP10-11*). Intriguingly, there was a higher proportion of *BRCA2* amplifications in tumors from men with African ancestry (21%; 3/14) than in tumors from men with European ancestry (1.1%; 4/346). This may account for the larger number of CNAs in this cohort of samples.

There were 148 genes in which the proportion of tumors with an amplification differed between men with East Asian ancestry compared with tumors from men with European ancestry (two-sided proportion test; Q < 0.05; Figure 2D). This includes a region on chromosome 7 spanning 26 genes that was amplified in 88% (9/11) of tumors of East Asian ancestry in our cohort. These data provide us with evidence that genes are mutated at different frequencies in individuals of different ancestry, providing further strong support to the idea that germline variation is essential for understanding the emergence of somatic phenotypes [24].

## 4. Discussion

Increasing evidence now shows that germline genetic variation strongly shapes the somatic profiles and evolutionary history of prostate cancer. Both rare deleterious variants in DNA damage response (DDR) genes and common polymorphisms have been shown to do so in Caucasian populations [2,24]. Recent sequencing studies have shown that tumors arising in men of Asian, African, or African-American ancestry show distinct somatic mutational features [25,26,27,28]. This study expands on these observations by examining a multi-ancestric cohort and provides the first analysis of CNAs in men of Middle Eastern descent.

Acknowledging the small number of samples in our Middle Eastern cohort, we were able to observe some key findings. Tumors in men of Middle Eastern ancestry showed less genomic instability than those of other ancestry groups, and harbored a larger proportion of amplifications in the glutathione S-transferase family of genes on chromosome 1 (*GSTM1*, *GSTM2*, *GSTM5*), the IQ motif-containing family of genes on chromosome 3 (*IQCF1*, *IQCF2*, *IQCF13*, *IQCF4*, *IQCF5*, *IQCF6*) and *PARP3*, an enzyme which is required for DNA repair not commonly mutated in tumors arising in men of other ancestries. *GSTM1* amplifications have been reported to be marginally associated with prostate cancer risk in a Caribbean population of African descent [29], warranting further investigation into the function of this gene in non-European cohorts. Interestingly, all three samples from men of Middle Eastern ancestry that had a high proportion of African admixture components had an amplification in *GSTM1*. Since our cohort lacked matching normal tissue, we were unable to determine if any of these mutations were germline CNAs. It is possible that our genomic analysis included germline CNAs, which would also be of significant interest. 

Using FISH, we identified a significant difference in *NKX3-1* deletions and *MYC* amplifications between tumors in men with European ancestry and men with Middle Eastern ancestry. These results are of significance given that *NKX3-1* is a gene known to be related to prostate epithelium development and its loss has been implicated in prostate cancer progression, and because *MYC* has been shown to have a role in aggressive prostate cancer. This data provides further evidence that the genomic backgrounds of tumors can differ between ethnicities and affect the evolutionary path of the tumor.

In previous work, we investigated the significance and incidence of key genetic alterations related to *ERG*, *PTEN,* and *SPINK1* in a different Middle Eastern cohort relative to a different European cohort [9]. The study documented that *ERG*, *PTEN,* and *SPINK1* genomic alterations occurred less frequently in the Middle Eastern cohort and that an enrichment of *ERG* and *PTEN* loss was not observed. Additionally, tumors in patients with Middle Eastern ancestry that had both a *PTEN* loss and *ERG* fusion had a higher risk of biochemical relapse when compared to the European population, where *PTEN* loss alone was associated with worse clinical outcome. These data confirm that the genomics of prostate cancer differs between men with Middle Eastern ancestry and those from East Asian, African, and European populations, which is likely reflective of factors leading to the higher incidence, rate of disease progression, and cause-specific mortality observed in other ancestries compared with men with Middle Eastern ancestry [30,31].

## 5. Conclusions

In conclusion, tumors arising in men of ME ancestry had fewer CNAs and lower PGA than those in men of other ancestries. There was no difference in the proportion of global CNA subtypes between the different ancestry groups, although there were differences in the proportions of CNAs in specific genes. Therefore, ancestry should be considered when investigating and characterizing biomarkers and molecular signatures relative to disease progression, prognosis, and potentially therapeutic targeting. Despite our limited sample size, our results document significant differences in key genes associated with prostate cancer progression and aggressiveness. The data also suggest that germline–somatic interactions may be at play in tumors arising in men of different ancestries. Further multi-omic studies of tumors arising in men of different ancestry are clearly warranted.

## Figures and Tables

**Figure 1 cancers-13-02363-f001:**
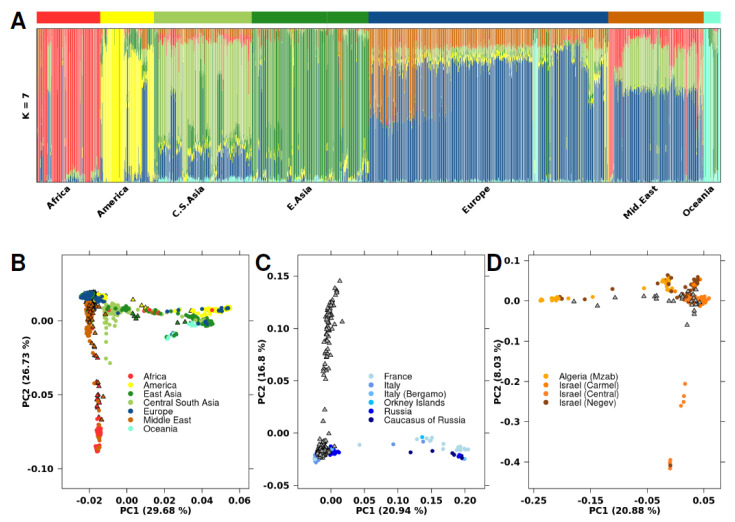
Population analysis with HGDP-CEPH SNP data. (**A**) ADMIXTURE cluster plot with *k* = 7. Each sample is represented by a vertical line partitioned into colored segments whose length is proportional to the ancestry coefficient in up to seven inferred populations. The cohort covariate bar represents which dataset the sample came from, where black indicates the current cohort and white indicates HGDP-CEPH. (**B**) PCA plot with 63,320 SNPs analyzing the same population as in Figure 1A. Samples are colored based on their assigned genetic ancestry, with circles indicating samples from the HGDP-CEPH dataset and triangles representing samples in the current cohort. (**C**) PCA plot including samples with European genetic ancestry. Samples are colored based on their geographic region within Europe, or grey if region is unknown. Circles indicate samples from the HGDP-CEPH dataset. Grey triangles represent samples with European ancestry in the current cohort; specific region is unknown. (**D**) PCA plot including samples with Middle Eastern ancestry. Samples are colored based on their geographic region within the Middle East, or grey if region is unknown. Circles indicate samples from the HGDP-CEPH dataset. Grey triangles represent the Middle Eastern samples in the current cohort; the specific region is unknown.

**Figure 2 cancers-13-02363-f002:**
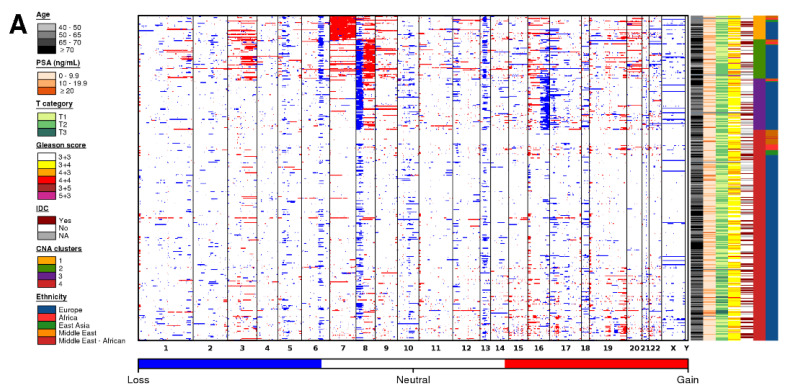
Copy number aberration profiles differ based on ancestry. (**A**) CNA profiles of tumors in men with Middle Eastern, African, East Asian, and European ancestry. Within the heatmap, red indicates a copy number amplification, while blue indicates a copy number deletion. Each row represents a tumor (ordered by subtype), and each column represents a gene ordered by genomic coordinate. Clinical features per tumor are shown in the covariate bar on the right, including the four previously identified CNA subtypes [11]. IDC; intraductal carcinoma or cribriform architecture. (**B**) Percent genome altered (PGA) differed across tumors from men with different ancestries. *p* value is from a Mann-Whitney test. (**C**) The proportion of tumors assigned to each subtype [11] did not differ between tumors from men with different ancestry. *p* value is from a Pearson’s chi-squared test. The colors of the stacked bars correspond to the CNA subtypes in Figure 1A. (**D**) Specific genes differed in their proportion between tumors from men with Middle Eastern, African, East Asian, and European ancestry. Genes are ordered by genomic coordinate per chromosome. Amplifications are represented in red, and deletions are represented in blue. Only significant Q values from proportion tests are shown comparing men with Middle Eastern (dark orange), African (red), or East Asian (green) ancestry to men with European ancestry.

**Table 1 cancers-13-02363-t001:** Summary of clinical features per ethnicity group.

Dataset	Europe	Africa	E. Asia	N. American	Mid. East
ISUP Grade Group					
1	60	1	3	1	9
2	200	9	6	3	12
3	75	4	2	1	3
4	7	0	0	0	1
5	2	0	0	0	0
NA	2	0	0	0	0
Age at Treatment					
40–50	7	1	0	1	0
50–65	160	4	5	3	8
65–70	74	2	1	0	12
≥70	105	7	5	1	5
Pre-treatment PSA					
≤9.9 ng/mL	4	1	11	3	0
10–19.9 ng/mL	93	3	0	1	8
≥20 ng/mL	249	10	0	1	17
T category					
T1	165	7	4	4	1
T2	181	7	7	1	13
T3	0	0	0	0	11
Intraductal carcinoma or cribriform architecture					
Yes	196	3	0	0	21
No	100	10	9	4	4
NA	50	1	2	1	0
Total	346	14	11	5	25

## Data Availability

The publicly available OncoScan SNP array data can be found on EGA under the accession EGAS00001002367. The Middle Eastern OncoScan SNP array data are being made available as well on EGA.

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
