# Peer review of "Copy Number Profiles of Prostate Cancer in Men of Middle Eastern Ancestry"

_cancers, 2021, doi:10.3390/cancers13102363_

Round 1
Reviewer 1 Report
Authors have address all the concerns made by the reviewers. They have also responded and justified comments related with the limitations of the study. Thus, I would accept the manuscript in the present form.
Reviewer 2 Report
Authors have addressed the requested comments
This manuscript is a resubmission of an earlier submission. The following is a list of the peer review reports and author responses from that submission.
Round 1
Reviewer 1 Report
The article by Albawardi et al. deals with the analysis of some genomic differences between Middle Eastern, African, East Asian and European populations to determine whether these differences could be responsible for the relatively better prognosis of Middle Eastern prostate cancer patients compared with the other groups. The study consists in the analyses of NKX3-1 deletions and MYC amplifications by fluorescent in situ hybridization of 42 tissue sections and SNP microarrays of 23 samples. Furthermore, the authors perform the analysis of gene level copy number aberrations (CNAs) from a dataset of 401 prostate cancer patients. The results showed a reduction of NKX3-1 deletion in Middle Eastern population compared with European ancestry. Moreover, the authors observed a reduction in CNAs in tumors from Middle Easter ancestry compared to the other ancestries.
In my opinion, the topic is important and interesting because address how the molecular profiles of prostate cancer may be related with prostate cancer prognosis and that these molecular signatures can vary depending on the population. Although the observation is of interest, this study appears to be too previous and with low statistical power to support the conclusions.
I would like to point out some concerns in more detail:
1. The title of the manuscript should be concise and reflect the conclusion of the study. Molecular Profiles is general and vague, as well as, identifies key differences. The main molecular difference is a decrease of NKX3-1 deletions and CNAs in Middle Eastern ancestry compared with Western populations so title should accordingly be rewritten.
2. The conclusion of the abstract “The lower number of somatic CNAs observed in prostate cancer arising in men of ME ancestry may explain the relatively good prognosis in this population” is not supported by the results showed in this article. It must be rewritten to be more accurate.
3. The definition of the cohort used in each analysis is not clear and it is very important to determine if there are risk subgroups that could influence the results. For example it would be very useful to include a column in the Supplementary Table S1 indicating which of the samples where used for the FISH or for the SNP microarray analyses.
If authors want to relate lower number of somatic CNAs in Middle Eastern ancestry with better prognosis of this population, it would be very interesting to include recurrence parameters, 5-year relapse-free survival data and type of treatment (surgery, radiotherapy, hormonal etc) in the Supplementary Table. It would help to be sure differences are due to the ancestry and not to other factors.
From the 42 tissues tested by FISH, only 22 were confirmed by OncoSacn SNP microarray profiling. Additionally, three other samples that had not been analyzed by FISH were included in the SNP profiling. The manuscript does not allow to identify the 22 samples and does not explain the reason why the other 20 samples were not tested. It cannot be proved if there is a bias in the samples selected.
A sample size estimation should be considered before the analyses: Accepting an α risk of 0.05 and a β risk of 0.2 in a two-sided test and assuming a RSD of 50%, 99 subjects are necessary per group to recognize as statistically significant a difference greater than or equal to 20%. In case of FISH analyses, the authors compare the results obtained from 346 individuals of the European ancestry with 25 individuals from Middle Eastern ancestry. If it is possible, the latter group should be increased to reach statistical significance.
The same limitation occurs when the authors compare somatic CNA profiles between ancestry groups having European ancestry 346 subjects, East Asian 11 subjects, African ancestry 14 subjects and Middle Eastern ancestry 25 subjects.
I am fully aware of the difficulty of increasing number of samples but if it is not possible, the authors should acknowledge the sample-size limitation of the study clearly along the manuscript and not only in a final sentence in the conclusions paragraph. I would appreciate a better explanation about the statistical test of each assay (parametric or non-parametric tests).
4. I would recommend clarifying if these samples are new or had been previously analyzed in your previous published work Abdelsalam RA et al (reference 7a). If cohort sample is different, could you please mention the novelty of this manuscript and explain the results obtained for ERG, PTEN and SPINK1?.
5. The authors should include in the discussion part the role of genes that they have found altered in ME samples. For example the important role of NKX3-1 homeobox gene in normal differentiation of the prostatic epithelium; its loss of function is an initiating event in prostate carcinogenesis.
6. The conclusions of the manuscript are general and unspecific. They should be rewritten indicating the real conclusion of the work and not speculating what has not been proven by your experimental work.
7.Other minor aspects mentioned below should also be addressed:
- The Funding of the study should be at the end of the manuscript, so line 105-106 of the paper should be erased.
- The use of 3a and 7a in the references section is confusing. What is the reason for this nomenclature? These references may be renumbered in increasing order for better understanding. Some formal aspect of the references should be revised (as example leave blank space after the number)
- Column D in Table S1 all PSA levels should contain same number of decimals.
- Line 146 indicates (Table 1; Table S1) and there is not Table 1 in the manuscript, are both the same table?
- In the supplementary Materials description (lines 304-326), all Figures and Tables words should be in bold.
- Line 54. Dot before Similarly should be removed.
Author Response
Reviewer 1:
The article by Albawardi et al. deals with the analysis of some genomic differences between Middle Eastern, African, East Asian and European populations to determine whether these differences could be responsible for the relatively better prognosis of Middle Eastern prostate cancer patients compared with the other groups. The study consists in the analyses of NKX3-1 deletions and MYC amplifications by fluorescent in situ hybridization of 42 tissue sections and SNP microarrays of 23 samples. Furthermore, the authors perform the analysis of gene level copy number aberrations (CNAs) from a dataset of 401 prostate cancer patients. The results showed a reduction of NKX3-1 deletion in Middle Eastern population compared with European ancestry. Moreover, the authors observed a reduction in CNAs in tumors from Middle Easter ancestry compared to the other ancestries.
In my opinion, the topic is important and interesting because address how the molecular profiles of prostate cancer may be related with prostate cancer prognosis and that these molecular signatures can vary depending on the population. Although the observation is of interest, this study appears to be too previous and with low statistical power to support the conclusions.
I would like to point out some concerns in more detail:
- The title of the manuscript should be concise and reflect the conclusion of the study. Molecular Profiles is general and vague, as well as, identifies key differences. The main molecular difference is a decrease of NKX3-1 deletions and CNAs in Middle Eastern ancestry compared with Western populations so title should accordingly be rewritten.
Response: We have changed the title of the manuscript to Copy Number Profiles of Prostate Cancer in Men of Middle Eastern Ancestry
- The conclusion of the abstract “The lower number of somatic CNAs observed in prostate cancer arising in men of ME ancestry may explain the relatively good prognosis in this population” is not supported by the results showed in this article. It must be rewritten to be more accurate.
Response: We have changed this sentence to ‘The lower number of somatic CNAs, specifically NKX3-1 and MYC, observed in prostate cancer arising in men of ME ancestry are of specific interest. Larger studies investigating ME populations are warranted to confirm these observations.
- The definition of the cohort used in each analysis is not clear and it is very important to determine if there are risk subgroups that could influence the results. For example it would be very useful to include a column in the Supplementary Table S1 indicating which of the samples where used for the FISH or for the SNP microarray analyses.
Response: We have added the following columns to Supplementary Table 1 to make the cohort information clearer; ‘NKX3-1 FISH status’, ’MYC FISH status’, ‘Has SNP array’. We have also compared the ME samples that have FISH data vs those that do not. There were no differences between T-category (p = 0.40; Pearson’s chi-squared test), ISUP Grade (p = 0.28; Pearson’s chi-squared test), IDC or cribriform status (p = 1, Pearson’s chi-squared test) or pre-treatment PSA (p = 0.48; Mann-Whitney U-test).
If authors want to relate lower number of somatic CNAs in Middle Eastern ancestry with better prognosis of this population, it would be very interesting to include recurrence parameters, 5-year relapse-free survival data and type of treatment (surgery, radiotherapy, hormonal etc.) in the Supplementary Table. It would help to be sure differences are due to the ancestry and not to other factors.
Response: We have added the type of treatment for each patient to Supplementary Table 1.
Data related to biochemical recurrence was not available. We carried out statistical analysis in relation to Gleason Grade groups and pathological features.
From the 42 tissues tested by FISH, only 22 were confirmed by OncoSacn SNP microarray profiling. Additionally, three other samples that had not been analyzed by FISH were included in the SNP profiling. The manuscript does not allow to identify the 22 samples and does not explain the reason why the other 20 samples were not tested. It cannot be proved if there is a bias in the samples selected.
Response: We have added the clinical information for the 20 samples with only FISH data to Supplementary Table 1 and they have been identified by the ‘Has SNP array’ column. As per the previous comment, we tested for differences between those samples that had FISH done and those that did not and there were no significant differences between the groups. Due to technicalities in the TMA construction and FISH experiments, not all FISH experiments were successful.
A sample size estimation should be considered before the analyses: Accepting an α risk of 0.05 and a β risk of 0.2 in a two-sided test and assuming a RSD of 50%, 99 subjects are necessary per group to recognize as statistically significant a difference greater than or equal to 20%. In case of FISH analyses, the authors compare the results obtained from 346 individuals of the European ancestry with 25 individuals from Middle Eastern ancestry. If it is possible, the latter group should be increased to reach statistical significance.
Response: We recognize this issue. However, as ME samples with good pathological/clinical data are difficult to obtain, we were not able to increase sample size for FISH currently, as this would require creating of additional TMAs and obtaining samples from additional patients. We truly appreciate the reviewer feedback and are working on trying to access larger cohorts for validation. We fully agree that it is likely that many sub-threshold hits remain (i.e. that we have a significant false negative rate), however that does not impinge upon the observations made in this study which represent a lower-bound on ME-associated prostate cancer CNAs.
The same limitation occurs when the authors compare somatic CNA profiles between ancestry groups having European ancestry 346 subjects, East Asian 11 subjects, African ancestry 14 subjects and Middle Eastern ancestry 25 subjects.
Response: Due to our sample-size, ancestry associations for mutation types are weakly-powered, highlighting the need for even larger studies. To determine the power of our ranked tests, we used the power.ladesign function from the clinfun R package (v1.0.15). Statistical power of each mutational density measure was calculated using the k-sample rank test under the Lehmann alternative hypothesis, where the relative average for each ethnicity group was in relation to those of European ancestry. The plot below shows the power vs sample size per group. The grey line indicates the sample size of the Middle Eastern cohort (n = 25).
I am fully aware of the difficulty of increasing number of samples but if it is not possible, the authors should acknowledge the sample-size limitation of the study clearly along the manuscript and not only in a final sentence in the conclusions paragraph. I would appreciate a better explanation about the statistical test of each assay (parametric or non-parametric tests).
Response: We added this sentence at the beginning of discussion “Acknowledging the small number of samples in our ME cohort, we were able to observe some key findings”. The specific tests that were performed are stated within the manuscript. For comparison of the groups, non-parametric tests (i.e. Mann-Whitney test) were used.
- I would recommend clarifying if these samples are new or had been previously analyzed in your previous published work Abdelsalam RA et al (reference 7a). If cohort sample is different, could you please mention the novelty of this manuscript and explain the results obtained for ERG, PTEN and SPINK1?.
Response: We added the following paragraph in discussion “In previous work, we investigated the significance and incidence of key genetic alterations related to ERG, PTEN and SPINK1 in a different Middle Eastern cohort relative to a different European cohort. The study documented that ERG, PTEN and SPINK1 genomic alterations occurred less frequently in the Middle Eastern cohort and that an enrichment of ERG and PTEN loss was not observed. Additionally, tumors from patients with Middle Eastern ancestry that had both a PTEN loss and ERG fusion had a higher risk of biochemical relapse when compared to the European population, where PTEN loss alone was associated with worst clinical outcome. This data confirms that the genomics of prostate cancer in men with Middle Eastern ancestry is different than those from East Asian, African and European populations, likely reflective of factors leading to the higher incidence, rate of disease progression and patients’ cause specific mortality observed in other ancestries compared to men with Middle Eastern ancestry”.
- The authors should include in the discussion part the role of genes that they have found altered in ME samples. For example the important role of NKX3-1 homeobox gene in normal differentiation of the prostatic epithelium; its loss of function is an initiating event in prostate carcinogenesis.
Response: We added the following in discussion “Using FISH, we identified a significant difference in NKX3-1 deletions and MYC amplifications between tumors from men with European ancestry and men with Middle Eastern ancestry. These results are of significance given that NKX3-1 is a gene known to be related to prostate epithelium development and its loss has been implicated in prostate cancer progression, and because MYC has been shown to have a role in aggressive prostate cancer. This data is further evidence that the genomic backgrounds of tumors can differ between ethnicities, and effect the evolutionary path of the tumor.” This may explain the hypothesized better prognosis or lower PCA incidence in ME population.
- The conclusions of the manuscript are general and unspecific. They should be rewritten indicating the real conclusion of the work and not speculating what has not been proven by your experimental work.
Response: We have added a ‘Conclusions’ section, which has been rewritten to include the conclusions that can be drawn from the study. Specifically, it says “In conclusion, we find that tumors arising in men of ME ancestry had fewer CNAs and lower PGA, than those of other ancestries. There was no difference in the proportion of CNA subtypes, based on the entire CNA profile, between the different ancestry groups, although there were differences in the proportion of CNAs in specific genes.”
7.Other minor aspects mentioned below should also be addressed:
- The Funding of the study should be at the end of the manuscript, so line 105-106 of the paper should be erased.
Response: Funding is noted after the discussion at the end of the manuscript.
- The use of 3a and 7a in the references section is confusing. What is the reason for this nomenclature? These references may be renumbered in increasing order for better understanding. Some formal aspect of the references should be revised (as example leave blank space after the number)
Response: The references have been fixed to be numbered in increasing order
- Column D in Table S1 all PSA levels should contain same number of decimals.
Response: PSA levels are now all stated to two decimal points
- Line 146 indicates (Table 1; Table S1) and there is not Table 1 in the manuscript, are both the same table?
Response: Table 1 was included as a pdf during the submission process
- In the supplementary Materials description (lines 304-326), all Figures and Tables words should be in bold.
Response: Any reference to a Table or Figure has been bolded
- Line 54. Dot before Similarly should be removed.
Response: This has been fixed within the manuscript

Reviewer 2 Report
This is an interesting manuscript. I have only few comments:
- Information about patients' age, PSA, symptoms etc... should be provided
- I do not understand why authors used TURP and radical prostatectomy specimens. The paper is centered on prostate cancer and TURP is generally performed for BPH. Please provide information about patients. Gleason score, were procedure performed in metastatic patients as palliative care?
- Without information about the patients and tumors there is no way to understand the meaning of the manuscript
Author Response
Reviewer 2:
This is an interesting manuscript. I have only few comments:
- Information about patients' age, PSA, symptoms etc... should be provided
Response: This information is available as a summary in Table 1 and more detailed information is available in Supplementary Table 1
- I do not understand why authors used TURP and radical prostatectomy specimens. The paper is centered on prostate cancer and TURP is generally performed for BPH. Please provide information about patients. Gleason score, were procedure performed in metastatic patients as palliative care?
Response: The clinical information of the ME cohort is provided in Table 1. All samples were from radical prostatectomies, except one sample which is from a TURP. Given the rarity of such ME samples with high GG, we assessed mixed sample types. This sample did not show any atypical molecular features.
- Without information about the patients and tumors there is no way to understand the meaning of the manuscript
Response: We have included patient’s clinical information in Supplementary Table 1. We hope that this aids the reviewer in the interpretation of our results.

Reviewer 3 Report
Authors have analyses the genomic landscape of a mostly understudied group and compared with existing database of patients from different ancestries. They performed FISH analysis for NXK3.1, a tumor suppressor that is commonly lost in PCa during the early stages of the disease in most patients. Additionally that also studied changes in MYCN and CNA among the middle eastern population and compared it with other data from other ethnicities. They observe that these commonly altered genes and selections are not significantly affected among patients with ME ancestry. Furthermore, they observe a specific amplification of gene associated with chromosome 1 particularly the glutathione-S-transferase family to be altered among these patients. They conclude that patients from ME ancestry set of alterations are different from other commonly known alterations in PCa.
Overall, the study is very interesting and is exploring a relatively less known aspect of impact of ancestral difference in genetic makeup of prostate tumors, with respect to the ME populations. Few concerns are as follows:
- It will to good to include the genomic methylation analysis to supplement the current findings and to attribute whether the genomic differences are also shared at the level of epibenthic modifications as well.
- Was there any distinction between the stage of tumors and NKX3.1 changes among the studies sample? There are certain events during PCa progression that occur at different stages of tumor development.
- Few full stops and English language may need some edits at few places.
Author Response
Reviewer 3:
Authors have analyses the genomic landscape of a mostly understudied group and compared with existing database of patients from different ancestries. They performed FISH analysis for NXK3.1, a tumor suppressor that is commonly lost in PCa during the early stages of the disease in most patients. Additionally that also studied changes in MYCN and CNA among the middle eastern population and compared it with other data from other ethnicities. They observe that these commonly altered genes and selections are not significantly affected among patients with ME ancestry. Furthermore, they observe a specific amplification of gene associated with chromosome 1 particularly the glutathione-S-transferase family to be altered among these patients. They conclude that patients from ME ancestry set of alterations are different from other commonly known alterations in PCa.
Overall, the study is very interesting and is exploring a relatively less known aspect of impact of ancestral difference in genetic makeup of prostate tumors, with respect to the ME populations. Few concerns are as follows:
- It will be good to include the genomic methylation analysis to supplement the current findings and to attribute whether the genomic differences are also shared at the level of epibenthic modifications as well.
Response: We agree that it would be very interesting to investigate the epigenetics of our Middle Eastern cohort, having methylation profiling would require a different platform which is beyond the scope of CNA assessment we carried out.
- Was there any distinction between the stage of tumors and NKX3.1 changes among the studies sample? There are certain events during PCa progression that occur at different stages of tumor development.
Response: There was no association between T category and NKX3-1 status in the Middle Eastern cohort FISH data (p = 0.694; Pearson’s chi-squared test) or the SNP array data (p = 0.448; Pearson’s chi-squared test).
- Few full stops and English language may need some edits at few places.
Response: We have thoroughly updated the manuscript to improve the grammar and readability
